# The Link between Obesity and Inflammatory Markers in the Development of Type 2 Diabetes in Men of Black African and White European Ethnicity

**DOI:** 10.3390/nu12123796

**Published:** 2020-12-11

**Authors:** Olah Hakim, Oluwatoyosi Bello, Meera Ladwa, Janet L. Peacock, A. Margot Umpleby, Geoffrey Charles-Edwards, Stephanie A. Amiel, Louise M. Goff

**Affiliations:** 1Diabetes Research Group, Departments of Diabetes & Nutritional Sciences, School of Life Course Sciences, Faculty of Life Sciences & Medicine, King’s College London, Waterloo Campus, Franklin-Wilkins Building, Room 3.87, London SE1 9NH, UK; olah.hakim@kcl.ac.uk (O.H.); oluwatoyosi.bello@kcl.ac.uk (O.B.); meera.ladwa@kcl.ac.uk (M.L.); stephanie.amiel@kcl.ac.uk (S.A.A.); 2Department of Epidemiology, Geisel School of Medicine at Dartmouth, Dartmouth College, Hanover, NH 03755-1404, USA; janet.peacock@kcl.ac.uk; 3School of Population Health and Environmental Sciences, King’s College London, London SE1 7EH, UK; 4Faculty of Health and Medical Sciences, University of Surrey, Guildford GU2 7XH, UK; m.umpleby@surrey.ac.uk; 5Medical Physics, Guy’s & St Thomas’ NHS Foundation Trust, London SE1 7EH, UK; geoff.charles-edwards@kcl.ac.uk; 6School of Biomedical Engineering & Imaging Sciences, King’s College London, London SE1 7EH, UK

**Keywords:** visceral adipose tissue, deep subcutaneous adipose tissue, superficial subcutaneous adipose tissue, inflammation, ethnicity, African, type 2 diabetes

## Abstract

In this study, we aimed to assess ethnic differences in visceral (VAT), deep subcutaneous (dSAT), and superficial subcutaneous (sSAT) adipose tissue and their relationships with inflammatory markers between white European (WE) and black West African (BWA) men with normal glucose tolerance (NGT) and type 2 diabetes (T2D). Forty-two WE (23 NGT/19 T2D) and 43 BWA (23 NGT/20 T2D) men underwent assessment of plasma inflammatory markers using immunoassays alongside Dixon magnetic resonance imaging to quantify L4-5 VAT, dSAT and sSAT. Despite no ethnic differences in sSAT and dSAT, BWA men exhibited lower VAT (*p* = 0.002) and dSAT:sSAT (*p* = 0.047) than WE men. Adiponectin was inversely associated with sSAT in WE (*p* = 0.041) but positively associated in BWA (*p* = 0.031) men with T2D. Interleukin-6 (IL-6) was associated with VAT in WE but not in BWA men with NGT (WE: *p* = 0.009, BWA: *p* = 0.137) and T2D (WE: *p* = 0.070, BWA: *p* = 0.175). IL-6 was associated with dSAT in only WE men with NGT (WE: *p* = 0.030, BWA: *p* = 0.833). The only significant ethnicity interaction present was for the relationship between adiponectin and sSAT (*P*_interaction_ = 0.003). The favourable adipose tissue distribution and the weaker relationships between adiposity and inflammation in BWA men suggest that adipose tissue inflammation may play a lesser role in T2D in BWA than WE men.

## 1. Introduction

The global increase in type 2 diabetes (T2D) is concomitant with the rise in obesity; hence, the role of adipose tissue dysfunction in driving insulin resistance is an area of increasing interest in T2D research [1,2]. Populations of African descent suffer greater prevalence rates of T2D compared to white populations, which is not entirely explained by obesity or socioeconomic factors [3,4]. It is widely accepted that the regional distribution of adipose tissue is a stronger predictor of metabolic risk compared to measures of whole-body adiposity such as body mass index (BMI) [5]. The deposition of abdominal fat, particularly visceral adipose tissue (VAT), has long been related to insulin resistance and T2D while the increased ratio of subcutaneous adipose tissue (SAT) to VAT has been shown to have metabolically protective qualities [6,7,8].

The deposition of VAT is driven by the reduced expandability of SAT, which causes a “spill over” of triglycerides that are subsequently stored in the visceral cavity. Further investigation of the SAT depot shows it is composed of two distinct compartments, superficial SAT (sSAT) and deep SAT (dSAT), which separate a fascial plane. The sSAT is considered to be the primary storage compartment of excess energy and the metabolically safest fat storage depot; however, upon an excess increase in sSAT, there is a disproportionate increase in dSAT, which is morphologically and metabolically likened to VAT [9,10]. Indeed, like the VAT to SAT ratio, the increased ratio of dSAT to sSAT is also related to insulin resistance and T2D, indicating a reduced capacity of sSAT to expand [11,12]. Adipocytes isolated from dSAT are larger and more loosely organised with greater lipolytic activity and pro-inflammatory cytokine expression compared to those from the sSAT depot [9,13,14].

Chronic low-grade systemic inflammation has been proposed as an underlying mechanism that mediates insulin resistance, linking dysfunctional adipose tissue with the development of T2D [15,16]. Characteristic of dysfunctional adipose tissue, both VAT and dSAT, have been shown to secrete greater pro-inflammatory cytokines, including tumour necrosis factor-α (TNF- α) and interleukin-6 (IL-6), compared to sSAT, which secretes greater adiponectin, an anti-inflammatory insulin sensitizer [17,18]. Previous studies have confirmed that sSAT has the lowest secretion of pro-inflammatory cytokines followed by dSAT then VAT, which has the strongest pathogenic potential with regard to cardiometabolic risk. Considering this, it is highly paradoxical that black populations suffer disproportionately from T2D since they have consistently been shown to have lower levels of VAT compared to their white counterparts [19,20]. Studies investigating the inflammatory contributions of SAT and VAT have suggested black populations have greater inflammatory markers than their white counterparts, which is particularly evident in black women [21]. However, after a weight loss intervention, black women showed a lesser decrease in their inflammatory markers compared to white women despite similar amounts of VAT and SAT loss [22]. This questions the role adipose tissue inflammation in the development of T2D in populations of African descent; however, studies in black individuals with T2D are limited. Therefore, we aimed to investigate the effect of ethnicity and glycaemic state on the relationships between VAT, dSAT and sSAT with specific markers of adipose tissue inflammation in men of white European (WE) and black West African (BWA) ethnicity with normal glucose tolerance (NGT) or T2D. Due to the greater prevalence of T2D in black populations, but lower levels of VAT, we hypothesised that there would be stronger relationships between measures of inflammation and VAT and dSAT in BWA men compared to WE men.

## 2. Materials and Methods

This investigation was conducted as part of the South London Diabetes and Ethnicity Phenotyping (Soul-Deep) study which is a cross-sectional study that aims to understand ethnic differences in the pathophysiology of T2D between WE and BWA men; the full protocol is published elsewhere [23]. This study was conducted between April 2013 and April 2019. Participants were recruited into the Clinical Research Facility at King’s College London for a screening assessment to assess eligibility for participation. All participants provided written informed consent prior to participation. The Soul-Deep study was approved by the London Bridge National Research Ethics Committee (15/LO/1121 and 12/LO/1859).

## 3. Participants

Recruitment of participants was conducted by advertisements in local newspapers, general practices, Facebook and leaflet distribution. Potential participants underwent a screening assessment to assess eligibility which included a health screening questionnaire to report age, participants’ self-declared ethnicity and that of their parents and grandparents, date of diabetes diagnosis if applicable, medical history, current medication and contraindications for magnetic resonance imaging (MRI). A fasting blood test measured full blood count, renal and liver function, HbA1c, lipid profile, sickle cell trait and auto-antibodies (anti-insulin, anti-GAD and anti-IA2). Anthropometric measurements were taken, which included height, weight, waist circumference (measured at the midpoint between the lowest rib and the iliac crest) and seated blood pressure. Men of WE or BWA ethnicity with either normal glucose tolerance (NGT) or T2D were eligible to take part in the study if they fulfilled the following criteria: (1) 18–65 years old;(2) had a BMI of 20–40 kg/m^2^ for NGT participants or 25–40 kg/m^2^ for T2D participants; (3) were of WE or BWA ethnicity as identified through birthplace of self, parents and grandparents; (4) had a HbA1c level <47.5 mmol/mol (6.5%) for NGT participants or ≤63.9 mmol/mol (8%) for T2D participants. The glycaemic status of the NGT participants was confirmed by a 2-h plasma glucose concentration <7.8 mmol/l during a 75 g oral glucose tolerance test that was conducted according to the procedures outlined by the World Health Organisation [24]. The glycaemic status of the participants with T2D was confirmed by a recent (less than 5 years) documented diagnosis of T2D. To minimise the effect of diabetes medication on the study outcomes, participants with T2D were only included if their T2D was being treated with metformin and/or lifestyle alone. Participants were excluded from the study if they had contraindications for MRI (such as metal implants or claustrophobia), were receiving treatment with thiazolidinedione, insulin, oral steroids, beta-blockers or any other medication that could affect the study outcomes or showed evidence of liver or kidney damage, determined from a serum alanine aminotransferase (ALT) level of 2.5-fold above the upper limit of the reference range or a serum creatinine level above 150 mmol/l, respectively, as well as if they tested positive for anti-insulin, anti-GAD or anti-IA2 auto-antibodies or had sickle cell disease.

## 4. Magnetic Resonance Imaging

To assess abdominal SAT (ASAT), VAT, dSAT and sSAT, participants underwent an MRI scan which took place at the radiology department at Guy’s Hospital, London, UK. Participants were instructed to attend the scan after an overnight fast. Additionally, participants on metformin therapy were instructed to cease taking it for 7 days prior to the scan. While lying in the supine position with surface coils placed on the scanned region, abdominal MRI data were acquired using a 2-point Dixon-based MRI sequence (repetition time: 6.77 ms; echo times: 4.77 ms (in-phase) and 2.39 ms (out-of-phase); flip angle: 10°) on a 1.5 T Siemens Aera scanner. To reduce motion artefacts, participants were instructed by the radiographer to hold their breaths for three bouts of 15 s while the images were acquired. From each participant, fat-only and water-only images were produced from the continuous abdominal axial T1-weighted gradient-echo images with a slice thickness of 3 mm. MRI data were analysed using the open-source image analysis software HOROS V 1.1.7 (www.horosproject.org; accessed 21/10/2017) by a single analyst who was blinded to clinical data. Areas of VAT, ASAT, dSAT and sSAT were determined from a single fat-only MRI image at the L4-5 anatomical position where the appropriate fat regions were manually highlighted to quantify their respective areas. dSAT and sSAT regions were distinguishable since they are separated by the fascia superficialis, which is visible in an axial MRI image. The intraclass correlation coefficients for VAT, ASAT, dSAT and sSAT between two independent observers were 0.981, 0.987, 0.982 and 0.936, respectively, which indicates high inter-rater reliability.

## 5. Biochemical Measurements

Fasting plasma adipokines and cytokines were determined using immunoassays (Affinity Biomarker Labs, London, UK). Plasma leptin and total adiponectin were measured using Human Quantikine enzyme-linked immunosorbent assay (ELISA) kits (Bio-Techne, Minneapolis, MN, USA); the mean intra- and inter-assay coefficients of variation (CV) were 3.2% and 4.4%, respectively, for leptin and 3.5% and 6.5%, respectively, for adiponectin. Plasma resistin, TNF-α, interferon-γ (IFN-γ), IL-6, IL-8, IL-10 and vascular endothelial growth factor (VEGF) were determined using a Human Proinflammatory multiplex immunoassay kit, Mesoscale Quickplex Discovery SQ120 (Meso Scale Discovery, Gaithersburg, MD, USA); mean intra- and inter-assay CV were <5.0% and 10.1%, respectively. Fasting plasma glucose was determined using an automated glucose analyser (Yellow Spring Instruments, 2300 STAT Glucose Analyser, Westlake, OH, USA); precision: ±2% of the reading or 2.5 mg/dL, whichever was larger.

## 6. Statistics

The study is part of the secondary outcomes of the Soul-Deep study which included 20 samples per group to allow the detection of a difference of one standard deviation with a power of 90% and 2-sided significance in the primary outcome variable (beta-cell insulin secretory function). Continuous variables were tested for normal distribution using a Shapiro–Wilk test and an assessment of the histograms. Skewed variables were log transformed to achieve a normal distribution prior to statistical analysis. Comparisons of continuous variables was conducted using a 2-way between–within groups analysis of variance (ANOVA) to test differences by ethnicity (WE and BWA) and glycaemic state (NGT and T2D), as well as their interaction (ethnicity*glycaemic state). Post-hoc analysis of variables that showed a significant ethnicity by glycaemic state interaction were conducted using independent sample t-tests. The strength of relationships between continuous variables were estimated using Pearson’s correlations or partial correlation when adjusting for age. Ethnic differences in the strength of relationships were examined by fitting a regression model with an interaction term for ethnicity. Data are presented as the mean with the standard deviation for normally distributed data or the geometric mean with the 95% confidence interval for log-transformed data. Statistical analyses were conducted with SPSS version 25.0 and *p* values < 0.05 were considered statistically significant.

## 7. Results

### 7.1. Characteristics

We studied 42 WE (23 NGT/19 T2D) and 43 BWA men (23 NGT/20 T2D); their clinical characteristics are presented in Table 1. All characteristics differed significantly by glycaemic state except for total cholesterol and high-density lipoprotein (HDL) cholesterol; this depicts the typical pattern of deterioration of several parameters in the T2D state. There were no significant ethnic differences in mean age, weight, BMI, HbA1c, fasting plasma glucose, blood pressure, low-density lipoprotein (LDL) and HDL cholesterol. Mean waist circumference, total cholesterol and triglycerides were significantly lower in the BWA men. There was no significant ethnicity by glycaemic state interaction for any characteristic.

### 7.2. Regional Adipose Tissue Deposition

Regional abdominal adipose tissue determined from MRI data are presented by ethnicity and glycaemic state in Table 1. All regional fat depots differed significantly by glycaemic state and were greater in the men with T2D. There were no ethnic differences in mean ASAT, dSAT and sSAT; however, mean VAT, VAT:ASAT and dSAT:sSAT were significantly lower in the BWA men. There was no significant ethnicity by glycaemic state interaction for any of the regional fat depots.

### 7.3. Inflammatory Markers

Levels of fasting plasma adipokines and inflammatory markers are presented by ethnicity and glycaemic state in Table 1. Mean adiponectin, leptin, IL-6, TNF-α, IL-8 and VEGF differed significantly by glycaemic state. There were no ethnic differences in leptin, IL-6, resistin, IFN-γ, IL-8 and VEGF; however, the BWA men had significantly lower mean adiponectin and TNF-α, and greater IL-10 compared to the WE men.

### 7.4. Relationships between Inflammatory Markers and Regional Adipose Depots

In the men with T2D, adiponectin showed an inverse association with sSAT in WE men but a positive association in BWA men (Figure 1); when modelled in a regression, a significant ethnicity interaction was present for this relationship (*P*_interaction_ = 0.003). Furthermore, in the combined NGT and T2D cohorts, adiponectin was inversely associated with sSAT and dSAT in the WE men (r = −0.34, *p* = 0.033 and r = −0.31, *p* = 0.052, respectively), but not BWA men (r = 0.13, *p* = 0.425 and r = −0.08, *p* = 0.606, respectively); these relationships remained significant after adjustment for age. In both the NGT and T2D states, leptin was significantly associated with VAT, dSAT and sSAT in the WE and BWA men, Figure 2. IL-6 showed positive associations with VAT in both WE and BWA men, but these relationships were significant in only the WE men (Figure 3). In the combined NGT and T2D cohorts, IL-6 was significantly associated with VAT in both WE and BWA men (r = 0.53, *p* = 0.001 and r = −0.34, *p* = 0.033, respectively); however, after adjustment for age, this relationship remained significant in only the WE men. Additionally, IL−6 was associated with dSAT in the WE men but not BWA men with NGT, Figure 3. TNF-α showed no significant associations with VAT, dSAT or sSAT in the WE and BWA men with NGT and T2D. However, when the NGT and T2D cohorts were combined, TNF-α was significantly associated with sSAT in the WE men (r = 0.32, *p* = 0.047) but not the BWA men (r = 0.07, *p* = 0.66), although this relationship diminished after adjustment for age.

## 8. Discussion

In this study, we explored the impact of ethnicity on regional adipose tissue deposition as well as the relationships between measures of regional adiposity and inflammatory markers in men of WE and BWA ethnicity in two glycaemic states, NGT and T2D. The main findings recognise ethnic distinctions in regional adipose tissue distribution, inflammatory markers and their relationships, whereby BWA men presented with a relatively favourable adiposity profile alongside lower pro-inflammatory cytokines but lower adiponectin. Furthermore, while VAT showed relationships with specific inflammatory markers in both WE and BWA men, sSAT showed relationships in only WE men indicating that there may be ethnic differences in the cytokine secretions of sSAT. Our data suggest that VAT may secrete pro-inflammatory cytokines in proportion to their size in WE and BWA men, while sSAT appears to be more pro-inflammatory, with increasing adiposity in WE men but not BWA men.

The ethnic differences in VAT and VAT:SAT that we found between the WE and BWA men supports previous findings which have consistently shown VAT is lower in black populations compared to their white counterparts with no ethnic differences in SAT [25,26]. Additionally, the lack of an ethnic difference in dSAT reported in our study is also similar to previous findings in adolescents, women and men [27,28,29]. However, unlike VAT and dSAT, studies comparing sSAT between black and white groups have reported contradicting findings, which may be due to differences in gender and obesity status of the subjects studied. In our study, we showed that sSAT did not differ between WE and BWA men, which has also been previously reported by Nazare et al. in a multi-ethnic study of men and women [30]; however, several studies have reported greater sSAT in black women and children compared to their white counterparts [27,28,29,31]. The studies that reported greater sSAT in black people were mainly conducted in obese populations and women, who generally have a greater capacity to store SAT compared to men, which may explain this disparity in the findings.

Due to their greater pathogenic characteristics, the VAT and dSAT depots show stronger relationships with insulin resistance than sSAT [10,32]; considering this, the lower VAT:SAT and dSAT:sSAT values in the BWA men suggest that the greater risk of T2D in black populations may not be explained by their body fat distribution. Previous investigations comparing VAT, dSAT and sSAT have revealed stronger relationships between several inflammatory markers with VAT and dSAT than sSAT, indicating that the cytokine secretions of VAT and dSAT may explain their associations with metabolic dysregulation [12,17,33]. Here we have explored whether the cytokine secretion of the various adipose depots may explain the greater risk of T2D in black men despite their relatively favourable adipose tissue distribution. We found that VAT showed associations with inflammatory markers in the WE and BWA men; however, dSAT and sSAT showed weaker associations in the BWA men than WE men.

Despite comparable levels of SAT, the BWA men exhibited lower levels of adiponectin compared to the WE men, which supports previous reports that have consistently shown adiponectin to be lower in black populations [34,35,36,37]. Since adiponectin is known to be an insulin sensitizer as well as an independent predictor of T2D, which decreases with increasing adiposity, it is expected that adiponectin would have a negative relationship with increasing SAT [38]. Indeed, in the WE men, adiponectin was negatively associated with sSAT; however, this was not the case in the BWA men; this suggests that there are ethnic differences in depot-specific adipose tissue dysfunction between black and white men where, with increasing adiposity, the sSAT depot may have a greater anti-inflammatory contribution in black men than white men. The metabolically favourable qualities of sSAT exist due to its lower pro-inflammatory secretions as well as its ability to store excess triglycerides efficiently, thereby preventing the spill-over of free fatty acids into ectopic fat depots [9]. Our suggestion of a protective role of sSAT during obesity in BWA men may explain the lower levels of VAT and intrahepatic fat that is typically reported in black populations [39,40,41].

The significant relationship between IL-6 and dSAT in the WE men but not BWA men suggests the inflammatory profile of dSAT may be greater in WE men than BWA. Our findings contrast those of Evans et al. who reported black women had a greater expression of several inflammatory genes isolated from adipose tissue compared to white women [28]; however, despite this, the inflammatory profile of black women only explained 20% of their variation in insulin sensitivity compared to 56% in white women, suggesting that the greater risk of T2D in black women may not be mediated by adipose tissue inflammation. To understand the inflammatory contributions of sSAT and dSAT and their role in T2D in black populations, further work should consider biopsy sampling of adipocytes to assess their cytokine gene expression and their associations with insulin sensitivity in black versus white men.

T2D is known to be an inflammatory state caused by the increased secretion of pro-inflammatory cytokines by macrophages that infiltrate the dysfunctional adipose tissue, a process which typically occurs during obesity [42]. Indeed, obesity is considered to be a state of low-grade chronic inflammation. Although it is not fully understood whether inflammatory cytokines are a cause of insulin resistance or are purely markers of dysfunctional adipose tissue, plasma levels of several cytokines including TNF-α, IL-6 and adiponectin are independent predictors of T2D [15,43]. Our data support the above notion since several pro-inflammatory markers including TNF-α and IL-6 were greater, while adiponectin was lower, in the T2D group compared to the NGT group. Considering the effect of ethnicity, there are some signs of a more favourable inflammatory profile in the BWA men, as indicated by the lower TNF-α as well as the greater levels of the anti-inflammatory cytokine IL-10. The lower TNF-α in the BWA men is supported by reports of Hyatt et al., who showed lower TNF-α in black women compared to white women, which they attributed to greater VAT in the white women since TNF-α was associated with VAT in white women but not black women [36]. However, we reported no associations between TNF-α and VAT in both ethnic groups.

Despite the lower VAT in BWA men than WE men, IL-6 was related to VAT in both WE and BWA men; however, these relationships were stronger in the WE men than BWA men in the NGT and T2D states. Overall, the weaker relationships between pro-inflammatory markers and adipose tissue depots in the BWA men contrasts with our hypothesis and indicates that adipose tissue inflammation during increased obesity may not explain the greater prevalence of T2D in black populations. Other characteristics of adipose tissue dysfunction that may explain the greater metabolic risk in BWA men that may be considered in future studies include genetic markers of adipogenesis, adipocyte morphology, and the influence of adipokines on ectopic fat deposition. Interestingly, recent studies propose that the reduced insulin clearance that is commonly reported in black populations may be a primary mechanism contributing to early hyperinsulinaemia and subsequent insulin resistance in the progression of T2D in black individuals [44,45]; this is, therefore, a hypothesis worthy of further investigation.

Our study has several strengths, which include the measurement of adipose tissue deposition using MRI, which is considered to be the gold standard non-invasive method for the quantification of body composition components. Our data are limited in that only one MRI image was used in the analysis of the various fat depots as opposed to multi-slice methods; however, L4-5 VAT and SAT areas show strong correlations with the respective volumes determined using consecutive MRI images from the whole abdominal cavity in both black and white men and women [46]. Moreover, our study does not consider measures of whole-body SAT or gluteal SAT, since the adipocytes isolated from gluteal SAT have been found to differ morphologically and metabolically from those isolated from abdominal SAT [47]. Another strength is the inclusion of only men since most previous studies focus on women due the greater prevalence of obesity in black women compared to black men. Furthermore, epidemiological studies have suggested that the development of T2D is driven by excess adiposity more so in black women than black men [48,49]; hence, future work should consider gender and ethnic differences to elucidate these speculations. The cross-sectional nature of our study and the lack of mechanistic analyses of the different adipose depots limit the implications of our findings. Additionally, the small sample size may have limited the reliability to detect ethnic differences, although our study is comparable in size to other studies conducted on inflammatory markers and adiposity in a black versus white population.

To conclude, our data suggest that ethnic differences exist in the inflammatory cytokine contribution of various adipose tissue depots between black and white men, which may impact the development of T2D. We have demonstrated that BWA men have relatively favourable adipose tissue distribution, as indicated by lower VAT and dSAT:sSAT as well as weaker relationships between adiposity and inflammatory markers compared to WE men; therefore, the degree of adipose tissue inflammation appears to be directly related to the level of adiposity more so in WE men than BWA men. Furthermore, adipose tissue distribution and adipose tissue inflammation may not explain the greater risk of T2D in black populations. To expand the understanding of inflammation in T2D in black ethnic groups, future studies should focus on exploring the mechanistic role of inflammatory markers in insulin resistance.

## Figures and Tables

**Figure 1 nutrients-12-03796-f001:**
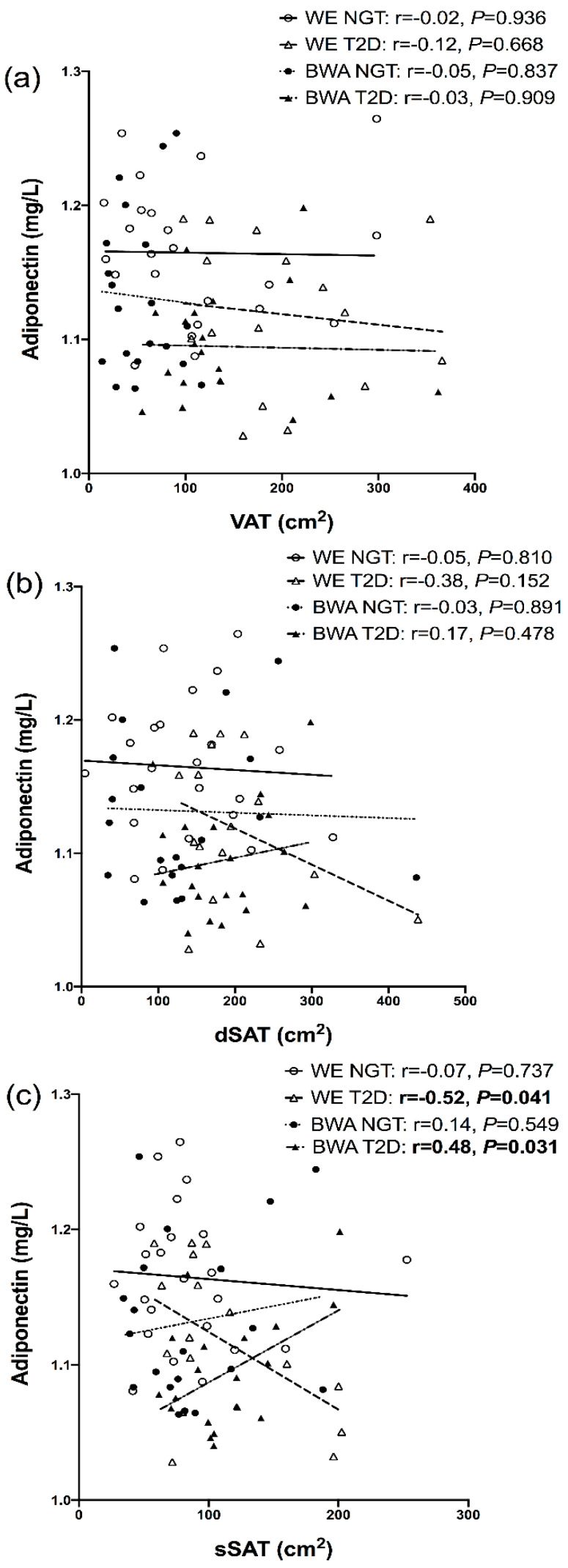
Relationships between adiponectin and visceral adipose tissue (VAT), (**a**), deep subcutaneous adipose tissue (dSAT), (**b**) and superficial SAT (dSAT), (**c**), in white European (WE) and black West African (BWA) men with normal glucose tolerance (NGT) and type 2 diabetes (T2D). Adiponectin underwent a +10 then log transformation to achieve a normal distribution in the positive axis. Bold *p*-values indicate significance of <0.05.

**Figure 2 nutrients-12-03796-f002:**
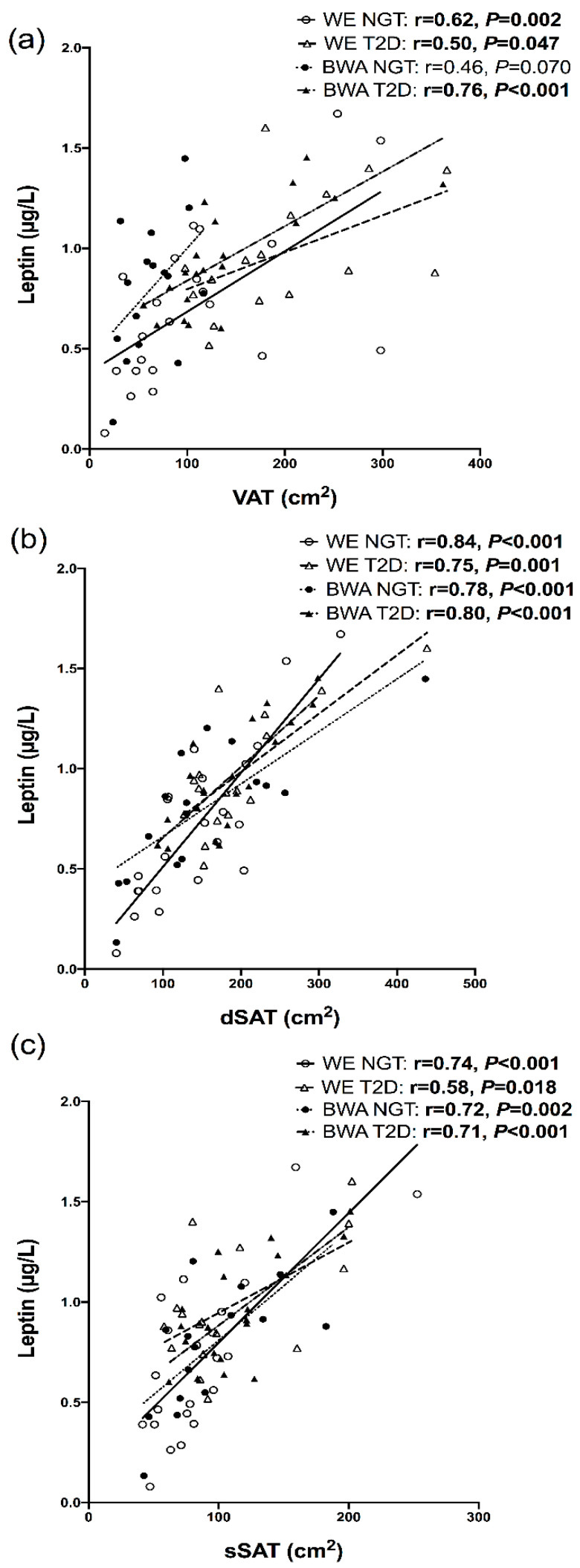
Relationships between leptin and visceral adipose tissue (VAT), (**a**), deep subcutaneous adipose tissue (dSAT), (**b**) and superficial SAT (dSAT), (**c**), in white European (WE) and black West African (BWA) men with normal glucose tolerance (NGT) and type 2 diabetes (T2D). Leptin was log transformed to achieve a normal distribution. Bold *p*-values indicate significance of <0.05.

**Figure 3 nutrients-12-03796-f003:**
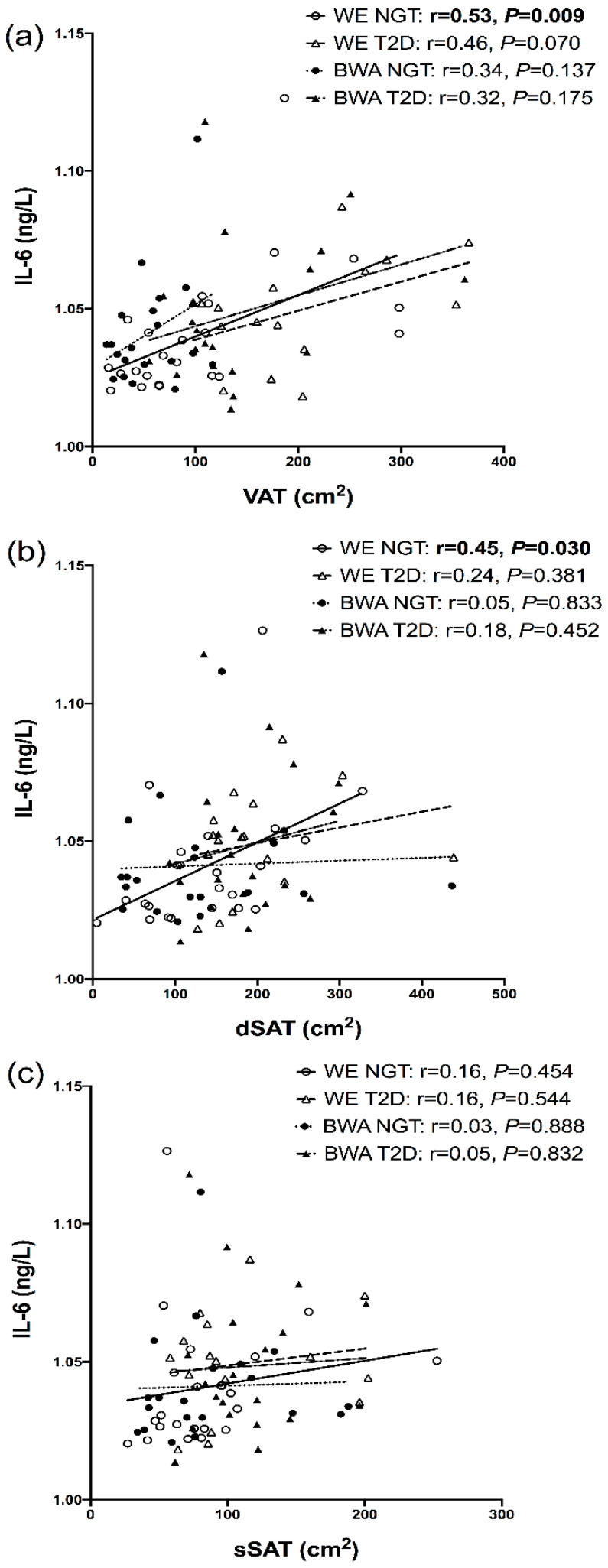
Relationships between interleukin-6 (IL-6) and visceral adipose tissue (VAT), (**a**) deep subcutaneous adipose tissue (dSAT), (**b**) and superficial SAT (dSAT), (**c**), in white European (WE) and black West African (BWA) men with normal glucose tolerance (NGT) and type 2 diabetes (T2D). IL-6 underwent a +10 then log transformation to achieve a normal distribution in the positive axis. Bold *p*-values indicate significance of <0.05.

**Table 1 nutrients-12-03796-t001:** Clinical characteristics, regional adipose tissue deposition and inflammatory markers in the white European (WE) and black West African (BWA) men with normal glucose tolerance (NGT) and type 2 diabetes (T2D).

	NGT		T2D				
	WE (*n* = 23)	BWA (*n* = 23)	WE (*n* = 19)	BWA (*n* = 20)	*P* eth	*P* gly	*P* eth*gly
Age (years) *	33.5 (28.4–39.5)	28.8 (24.6–33.7)	55.7 (52.3–59.3)	53.8 (50.2–57.8)	0.15	**<0.001**	0.36
Weight (kg)	86.5 ± 16.5	84.1 ± 14.6	100.6 ± 16.7	92.4 ± 11.8	0.11	**0.001**	0.38
BMI (kg/m^2^)	26.5 ± 4.5	26.7 ± 3.6	31.9 ± 4.3	30.0 ± 3.5	0.34	**<0.001**	0.24
Waist circumference (cm)	93.8 ± 14.6	87.5 ± 9.2	112.3 ± 12.7	104.9 ± 9.7	**0.009**	**<0.001**	0.83
HbA1c (%)	5.44 ± 0.24	5.54 ± 0.48	6.68 ± 0.74	6.76 ± 0.70	0.46	**<0.001**	0.94
HbA1c (mmol/mol)	35.9 ± 2.9	37.0 ± 5.27	49.5 ± 8.19	50.4 ± 7.80	0.45	**<0.001**	0.94
Fasting plasma glucose (mmol/l)	5.20 ± 0.39	5.13 ± 0.44	6.95 ± 1.35	6.71 ± 0.96	0.39	**<0.001**	0.63
Systolic BP (mm Hg)	121.9 ± 9.1	123.1 ± 12.3	132.1 ± 13.0	137.5 ± 13.7	0.21	**<0.001**	0.42
Diastolic BP (mm Hg)	71.1 ± 8.2	70.7 ± 11.5	82.6 ± 9.1	86.1 ± 7.4	0.45	**<0.001**	0.34
Total cholesterol (mmol/l)	4.76 ± 1.1	4.27 ± 1.1	4.41 ± 0.83	4.09 ± 0.70	0.051	0.19	0.68
LDL cholesterol (mmol/l)	2.99 ± 0.82	2.66 ± 0.87	2.34 ± 0.69	2.32 ± 0.53	0.27	**0.003**	0.35
HDL cholesterol (mmol/l)	1.27 ± 0.31	1.30 ± 0.42	1.22 ± 0.25	1.18 ± 0.37	0.97	0.23	0.62
Triglyceride (mmol/l) *	1.09 (0.86–1.33)	0.68 (0.57–0.79)	1.83 (1.41–2.26)	1.27 (0.94–1.61)	**<0.001**	**<0.001**	0.66
**Regional adipose tissue**							
VAT (cm^2^) *^,a^	79.0 (55–112)	46.1 (34–61)	184.4 (148–229)	128.1 (103–159)	**0.002**	**<0.001**	0.54
ASAT (cm^2^) *^,a^	193.2 (149–249)	181.9 (136–243)	291.0 (243–348)	285.6 (247–330)	0.73	**<0.001**	0.85
VAT:ASAT *^,a^	0.46 (0.34–0.59)	0.30 (0.20–0.39)	0.70 (0.52–0.87)	0.48 (0.39–0.58)	**0.002**	**<0.001**	0.70
dSAT (cm^2^) *^,a^	109.8 (76–159)	102.6 (73–145)	188.6 (159–223)	175.3 (150–205)	0.63	**<0.001**	0.99
sSAT (cm^2^) *^,a^	75.5 (62–93)	76.6 (60–97)	100.5 (81–125)	108.8 (93–127)	0.64	**0.002**	0.74
dSAT:sSAT ^a^	1.68 ± 0.84	1.42 ± 0.49	1.93 ± 0.47	1.65 ± 0.35	**0.047**	**0.075**	0.92
**Inflammatory markers**							
Adiponectin (mg/L) *	4.61 (3.89–5.37)	3.41 (2.64–4.22)	3.33 (2.46–4.26)	2.44 (1.88–3.02)	**0.005**	**0.003**	0.76
Leptin (µg/L) *^,b^	5.19 (3.44–7.83)	6.55 (4.42–9.70)	10.9 (7.7–15.5)	9.0 (6.8–12.0)	0.92	**0.003**	0.23
IL-6 (ng/L) *	0.99 (0.73–1.25)	0.95 (0.73–1.17)	1.32 (1.05–1.59)	1.18 (0.87–1.50)	0.49	**0.030**	0.71
TNF-α (ng/L) ^c^	2.48 ± 0.54	2.39 ± 0.75	3.03 ± 0.80	2.49 ± 0.55	**0.034**	**0.033**	0.13
IL-10 (ng/L) *^,d^	0.57 (0.44–0.70)	0.75 (0.62–0.88)	0.51 (0.41–0.61)	0.63 (0.49–0.80)	**0.019**	0.16	0.60
Resistin (µg/L) *^,e^	4.07 (3.57–4.63)	4.38 (3.68–5.21)	3.51 (3.00–4.11)	3.81 (3.13–4.64)	0.32	0.073	0.95
IFN-γ (ng/L) *^,f^	4.29 (3.62–5.07)	5.03 (4.04–6.26)	5.88 (4.7–7.3)	4.08 (2.6–6.4)	0.38	0.65	**0.027**
IL-8 (ng/L) ^g^	8.44 ± 3.62	8.15 ± 2.16	9.86 ± 4.50	11.0 ± 3.96	0.59	**0.008**	0.37
VEGF (ng/L) *	51.3 (37.6–70.0)	60.1 (43.4–83.2)	83.2 (62.5–110.7)	90.7 (65.5–125.5)	0.43	**0.004**	0.82

Data presented as mean ± SD or geometric mean (95% CI) for log transformed data (*****). *P* eth: main effect for ethnicity, *P* gly: main effect for glycaemic state, *P* eth*gly: interaction effect for ethnicity*glycaemic state determined using a 2-way between-groups ANOVA. Bold *p*-values indicate significance of <0.05. ^a^ N: NGT BWA = 20, T2D WE = 16; ^b^ N: NGT WE = 22, NGT BWA = 17; ^c^ N: T2D BWA = 18; ^d^ N: T2D BWA = 18; ^e^ N: NGT WE = 22; ^f^ N: NGT WE = 22, NGT BWA = 19, T2D WE = 18, T2D BWA = 11; ^g^ N: T2D BWA = 19. Abbreviations: ASAT, abdominal subcutaneous adipose tissue; BMI: body mass index; BP, blood pressure; dSAT, deep SAT; IFN, interferon; IL, interleukin; sSAT superficial SAT; TNF, tumor necrosis factor; VAT, visceral adipose tissue; VEGF, vascular endothelial growth factor.

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
