# Peer review of "The Link between Obesity and Inflammatory Markers in the Development of Type 2 Diabetes in Men of Black African and White European Ethnicity"

_nutrients, 2020, doi:10.3390/nu12123796_

Round 1
Reviewer 1 Report
The following comments may help improve the manuscript:
1. MR measurement were done by a single analyst only. Please get them done independently by another rater and determine intra-class correlation coefficient.
2. Merely a single MR slice was analysed. Please extend the measurements and determine abdominal volumes.
3. Validity and reproducibility of biochemical measurements was not reported.
4. The relationships between inflammatory markers and fat depots was investigated in an overly simplistic manner. At the very lease, all the analyses have to be adjusted for age and sex. There are certainly other covariates worth considering.
Reviewer 2 Report
This study examined the ethnic differences between European (WE) and black west African (BWA) between sSAT, dSAT, and VAT depots and their relationships with inflammatory markers between T2D and normal glucose tolerance men. The authors conclude that ethnic differences between inflammatory cytokine and various adipose tissues contributed to the development of type 2 diabetes in WE and BWA.
Globally considered this manuscript contains interesting information about ethnic differences, fat mass, and inflammation markers, however, I have some remarks to be considered.
- Authors stated in the introduction, that studies in the regional adipose deposition and markers of inflammation are limited. However, some studies showed these relationships in African Americans ( Joan F Carroll Obesity, 2009 Jul;17(7):1420-7, Gordon Fisher, Obesity 2012 Apr;20(4):715-20 etc..). Authors should restate why their work is different from these studies before presenting the goal of the study.
- Tables 2 and 3 are not fully clear. In Table 2 the authors showed pooled NGT and T2D subjects in the WE group compared to pool data in the BWA group. Since the characteristics of the T2D are very different than normal subjects, authors can discuss pool data in the discussion secession. It would be clearer if authors could present the significant correlations between the marker of inflammations including adiponectin, IL-6, and Leptin with sSAT, dSAT, and VAT as figures. Each figure can provide a marker of inflammation such as adiponectin correlations with sSAT, dSAT, and VAT in T2D and NGT in WE and BWA group. The authors may keep one table presenting non-significant data or other makers of inflammations that they measured such as resistin, IFN, IL-8, IL10 and VEGF. The data from these markers should be included in the paper.
- This study is limited measuring only visceral and subcutaneous fat (dSAT and sSAT) by one silence taken at L4-L5 position. A single L4-L5 MRI may not be comparable between BWA and WE, since the fat distribution is different between two ethnicities (Katzmarzyk PT et al. Am.J.Clin.Nutr2010; 91:7-15). The authors should discuss this point in the discussion.
- Authors state that (page 13 line 315) adipose tissue inflammation during increased obesity may not explain the greater prevalence of T2D in BWA. They suggested that adipose tissue dysfunction, gender markers of adipogenesis, adipocyte morphology, and the influence of adipokines on ectopic fat should be studied and may be important for the pathogenesis of T2D in BWA. The authors could include other important factors involved in the development of T2D such as fatty liver or insulin clearance in the discussion. Authors showed previously that low insulin clearance may be the primary mechanism of hyperinsulinemia in BWA. (Ladwa M Diabetes Obes Metab, 2020 Jun 2).
- The manuscript tends to overestimate the implications of the data to explain the inflammation markers and fat mass and the development of T2D in BWA and WE. Authors should state the study limitation of their study, the data provided are based on correlations and not mechanistic studies in the different adipose tissues.
Reviewer 3 Report
Here, the authors investigated the relationship between adipose tissue mass analyzed by MRI, ethnic differences, and inflammation associated with type 2 diabetes. The purpose is very interesting and valuable.
However, the age difference between the NGT group and the T2D group is too large. As is well known, age is a very important factor in considering the pathological condition of type 2 diabetes and obesity, especially the assessment of inflammation of adipose tissue. Therefore, the analysis between NGT and T2D is not appropriate. It is not surprising that there are significant differences (P-gly) in various characteristics between NGT and T2D. Rather, I am surprised that there is no significant difference between NGT and T2D in inflammatory markers. In addition, the number of n is too small to analyze the difference between WE and BWA. The author needs to rethink experimental design for the purpose.
Round 2
Reviewer 1 Report
No comments.
Author Response
No comments to respond to.
Reviewer 3 Report
I think this revision has been appropriately modified.
Author Response
Thank you for your approval of the amendments.